

# A closer look at severe acute kidney injury: risk factors and outcomes in PD-1/PD-L1 antibody treatment from a retrospective study

Yuemeng Wu[1,2], Lingfan Luo[2,3], Xin Sun[4], Xiaolan Ye[5], Yan Ren[2], Wei Zhang[2], Shuangshan Bu[1], Yiwen Li[2], Bin Zhu[2] and Lina Shao[2,3]

[1] Department of Nephrology, Dongyang Hospital (Affiliated Wenzhou Medical University), Jinhua, Zhejiang, China

[2] Urology & Nephrology Center, Department of Nephrology, Zhejiang Provincial People's Hospital (Affiliated People's Hospital), Hangzhou Medical College, Hangzhou, Zhejiang, China

[3] The Second Clinical College of Hangzhou Normal University (Zhejiang Provincial People's Hospital), Hangzhou, Zhejiang, China

[4] Department of Medical Oncology, Cancer Center, Key Laboratory of Tumor Molecular Diagnosis and Individualized Medicine of Zhejiang Province, Zhejiang Provincial People's Hospital (Affiliated People's Hospital, Hangzhou Medical College), Hangzhou, Zhejiang, China

[5] Center for Clinical Pharmacy, Cancer Center, Department of Pharmacy, Zhejiang Provincial People's Hospital (Affiliated People's Hospital, Hangzhou Medical College), Hangzhou, Zhejiang, China

Corresponding author
Lina Shao, shaolina@hmc.edu.cn

## ABSTRACT

**Background**. Immune checkpoint inhibitors (ICIs) have improved cancer survival but increase the risk of adverse events, including acute kidney injury (AKI). Severe AKI, though rare, can disrupt treatment and worsen outcomes. Yet, research on risk factors for severe AKI in patients on PD-1/PD-L1 therapies is limited. This study aimed to identify these risk factors.

**Methods**. This retrospective cohort study analyzed electronic medical records from Zhejiang Provincial People's Hospital from January 2019 to July 2023. In total, 907 patients who met the inclusion criteria, with a median age of 64 years, were included in the analysis. Least Absolute Shrinkage and Selection Operator (LASSO) and Cox regression analyses were conducted to determine independent risk factors for severe AKI.

**Results**. Severe AKI was observed in 3.2% of patients with AKI, with a significantly higher mortality rate than in non-AKI patients (20.7% *vs.* 4.1%) during the follow-up period. Multivariate Cox regression analysis identified elevated gamma-glutamyl transferase (hazard ratio (HR): 1.17), diuretic use (HR: 3.61), nonsteroidal anti-inflammatory drug (NSAID) use (HR: 4.58), and cytotoxic drugs (HR: 5.04) as independent risk factors for severe AKI. Only 11 patients (37.5%) with severe AKI recovered.

**Conclusions**. This study highlights the importance of monitoring these factors to reduce the risk of severe AKI in patients receiving PD-1/PD-L1 antibody therapy.

## INTRODUCTION

Immune checkpoint inhibitors (ICIs) have emerged as a vital component of cancer immunotherapy, markedly boosting prognosis and extending survival for individuals with diverse cancer diagnoses (*Perazella & Shirali, 2020*; *Sprangers et al., 2022*; *Yi et al., 2022*). These therapies work by inhibiting cytotoxic T lymphocyte-associated antigen-4 (CTLA-4) and programmed death-1 (PD-1) receptors or their ligand, programmed death ligand-1 (PD-L1), thereby boosting the body's antitumor immune response (*Barbir, Kitchlu & Herrmann, 2024*).

Despite their benefits, ICIs can induce a range of immune-related adverse events (irAEs) that affect multiple organs and tissues (*Postow, Sidlow & Hellmann, 2018*; *Quach et al., 2021*). Commonly affected systems include the skin, gastrointestinal tract, liver, and endocrine glands; however, ICIs can also impact renal function, potentially leading to acute kidney injury (AKI) (*Shingarev & Glezerman, 2019*; *Meraz-Muñoz et al., 2020*). AKI, marked by a rapid decrease in glomerular filtration rate (GFR) occurring within hours or days, can lead to rapid and severe health deterioration, with some cases resulting in irreversible loss of renal function (*Kellum, Norbert & Aspelin, 2012*). A meta-analysis of randomized control trials reported a 2.2% incidence of AKI among patients treated with PD-1/PD-L1 antibodies, which represents a four-fold increased risk compared to patients receiving non-nephrotoxic chemotherapy (*Manohar et al., 2019*). While the incidence of ICI-related AKI is relatively low, its consequences can be substantial, particularly for severe AKI, which can significantly disrupt cancer treatment plans and adversely affect patient outcomes (*García-Carro et al., 2022*).

Current research on ICI-related AKI suggests that most cases are mild and do not necessitate changes in cancer therapy per clinical guidelines. However, severe AKI typically requires medical intervention and adjustments to the antitumor regimen (*Schneider et al., 2021*), underscoring the importance of recognizing and managing this condition effectively. Prior studies have broadly examined risk factors for ICI-induced AKI broadly (*Seethapathy et al., 2019*; *Meraz-Muñoz et al., 2020*; *Ji et al., 2022*; *Liu et al., 2023a*; *Liu et al., 2023b*), but few have specifically investigated risk factors for severe AKI in patients with cancer undergoing PD-1/PD-L1 antibody therapy. This gap highlights the need for further exploration to identify these risk factors and better understand their implications. Such insights could enable earlier intervention and improved prognostic outcomes for patients at risk of severe AKI. Hence, this study aimed to identify risk factors associated with severe AKI in patients treated with PD-1/PD-L1 antibody therapy.

Specifically, we found that gamma-glutamyl transferase (GGT), diuretic use, nonsteroidal anti-inflammatory drug (NSAID) use, and cytotoxic drug use were associated with an increased risk of severe AKI in patients receiving PD-1/PD-L1 antibody therapy.

## MATERIALS & METHODS

### Study design

This retrospective cohort study analyzed patients who received PD-1/PD-L1 antibody therapy at Zhejiang Provincial People's Hospital between January 2019 and July 2023 using

data from the hospital's electronic medical record system. The patients were categorized into two groups: those who developed severe AKI and those who did not, based on AKI occurrence after treatment with ICIs. Patients with cancer who received PD-1/PD-L1 antibody therapy one or more times. The exclusion criteria were (1) the absence of initial or subsequent measurements of creatinine levels, (2) an estimated glomerular filtration rate (eGFR) <60 mL/min/1.73 m$^2$, (3) age greater than 85 years, and (4) a follow-up period shorter than 7 days. Patients were followed up from the initiation of PD-1/PD-L1 therapy until the occurrence of severe AKI or death.

Since this was a retrospective study, the requirement for informed consent was waived. This study was approved by the Ethics Research Committee of the Zhejiang Provincial People's Hospital (approval number: QT2024180).

## Data collection

Data for each patient were collected using an electronic medical record system, including demographic information (sex, age), medical history (smoking, alcohol consumption), vital signs (blood pressure), comorbidities, extrarenal iRAEs, tumor type, drug combinations, and laboratory data. The recorded comorbidities included diabetes, cardiovascular disease, chronic obstructive pulmonary disease (COPD), and cirrhosis. Extrarenal iRAEs encompassed a range of conditions such as thyroid dysfunction, skin injuries, fever, pneumonia, and drug-induced liver injury. Tumor types were classified, including head and neck tumors, hepatobiliary and pancreatic system tumors, lung cancer, breast cancer, gastrointestinal tract tumors, urinary system tumors, reproductive system tumors, and melanoma. Combination drugs included therapeutic agents such as proton pump inhibitors (PPIs), renin-angiotensin-aldosterone system inhibitors (RAASis), nonsteroidal anti-inflammatory drugs (NSAIDs), nephrotoxic antibiotics, iohexol, H2 receptor antagonists, diuretics (*e.g.*, furosemide, spironolactone), and chemotherapy agents (*e.g.*, platinum-based compounds, gemcitabine, cyclophosphamide, and interleukin therapies). Nephrotoxic drugs include aminoglycosides, sulfonamides, and vancomycin. Laboratory data included markers of anemia, uric acid levels, liver function tests, serum albumin, lipid profiles, serum inorganic phosphate, inflammatory markers, urinary leukocytes, and urinary specific gravity.

AKI was defined within 12 months of the first administration of PD-1/PD-L1 antibodies, based on the internationally accepted Kidney Disease Improving Global Outcomes (KDIGO) standard, and staged according to the magnitude of the increase (*Kellum, Norbert & Aspelin, 2012*). The stages of AKI were classified as follows: stage 1, a 1.5 to 1.9-fold increase in serum creatinine (SCr) from baseline or an absolute increase of at least 26.5 µmol/L; Stage 2, a 2.0 to 2.9-fold increase in SCr from baseline; and Stage 3, either a 3.0-fold increase in SCr from baseline or an SCr level ≥353.6 µmol/L. Stages 2 and 3 were classified as severe AKI. The baseline creatinine level was the closest creatinine level before starting the first PD-1/PD-L1 antibody treatment.

## Statistical analysis

The baseline characteristics of the cohort were summarized using frequency counts (expressed as percentages) for categorical variables and the median (accompanied by

the interquartile range [IQR]) for continuous variables. To evaluate differences between groups, categorical variables were analyzed using the chi-squared test, or when appropriate, Fisher's exact test, whereas continuous variables were compared using the Wilcoxon test. When a variable had less than 20% missing values, multiple imputation by chained equations was performed using the 'mice' package in R (version 4.4.0; R Foundation for Statistical Computing, Vienna, Austria). This approach assumes that data were missing completely at random, allowing observed variables to inform the imputation process (*Li, Stuart & Allison, 2015*; *Sterne et al., 2009*). The Least Absolute Shrinkage and Selection Operator (LASSO) method was applied to identify significant variables and prevent model overfitting (*Li et al., 2022*). In this regression model, the coefficients are penalized based on the value of λ, the LASSO regularization parameter. A larger penalty shrinks the estimate of less important factors toward zero, leaving only the strongest predictors. Variables selected by the LASSO regression were then included in a Cox regression analysis to identify risk factors for developing severe AKI in those patients who received PD-1/PD-L1 antibodies. The variable with a $p$-value of <0.05 in the univariate COX analysis was selected as the candidate risk variable in the multivariate COX analysis. The combination aims to balance model parsimony and statistical power (*Tibshirani, 1997*). Survival differences of severe AKI according to different risk factors were visualized using Kaplan–Meier survival plots. Statistical analysis was performed using SPSS software (version 26.0; IBM Corp., Armonk, NY, USA) and R software (version 4.4.0; R Foundation for Statistical Computing, Vienna, Austria). R packages used for data analysis and visualization included 'ggplot2' (Version 3.5.1) for data visualization, 'survival' (Version 3.6-4) for survival analysis, 'glmnet' (Version 4.1-8) for LASSO regression, as well as 'mice' (Version 3.16.0), 'missForest' (Version 1.5), 'VIM' (Version 6.2.2), 'Hmisc' (Version 5.1-2), 'rms' (Version 6.8-0), 'regplot' (Version 1.1), 'tableone' (Version 0.13.2), 'survminer' (Version 0.4.9), and 'survivalROC' (Version 1.0.3.1). Statistical significance was at a $p$-value of <0.05 for all tests.

## RESULTS

### Characteristics of patients with severe AKI

This study included 1,840 patients treated with PD-1/PD-L1 antibodies between January 2019 and July 2023. Patients were excluded for the following reasons: 82 patients lacked baseline creatinine data, 431 had no follow-up creatinine measurements, 48 had a baseline eGFR <60 mL/min/1.73 m$^2$, 17 were over 85 years old, and 263 had follow-up periods <7 days (Fig. 1). After these exclusions, the final cohort comprised 907 patients, including 660 men (72.8%) and 247 women (27.2%) with a median age of 64 years (IQR: 58–72) and a median baseline eGFR of 103.14 mL/min/1.73 m$^2$ (IQR: 87.85–121.76). In this cohort, 29 patients (3.2%) developed severe AKI, with 23 cases (79.3%) classified as stage 2 and six cases (20.7%) as stage 3. Mortality was significantly higher among patients with severe AKI, with a death rate of 20.7%, compared to 4.1% in those without AKI ($p < 0.001$).

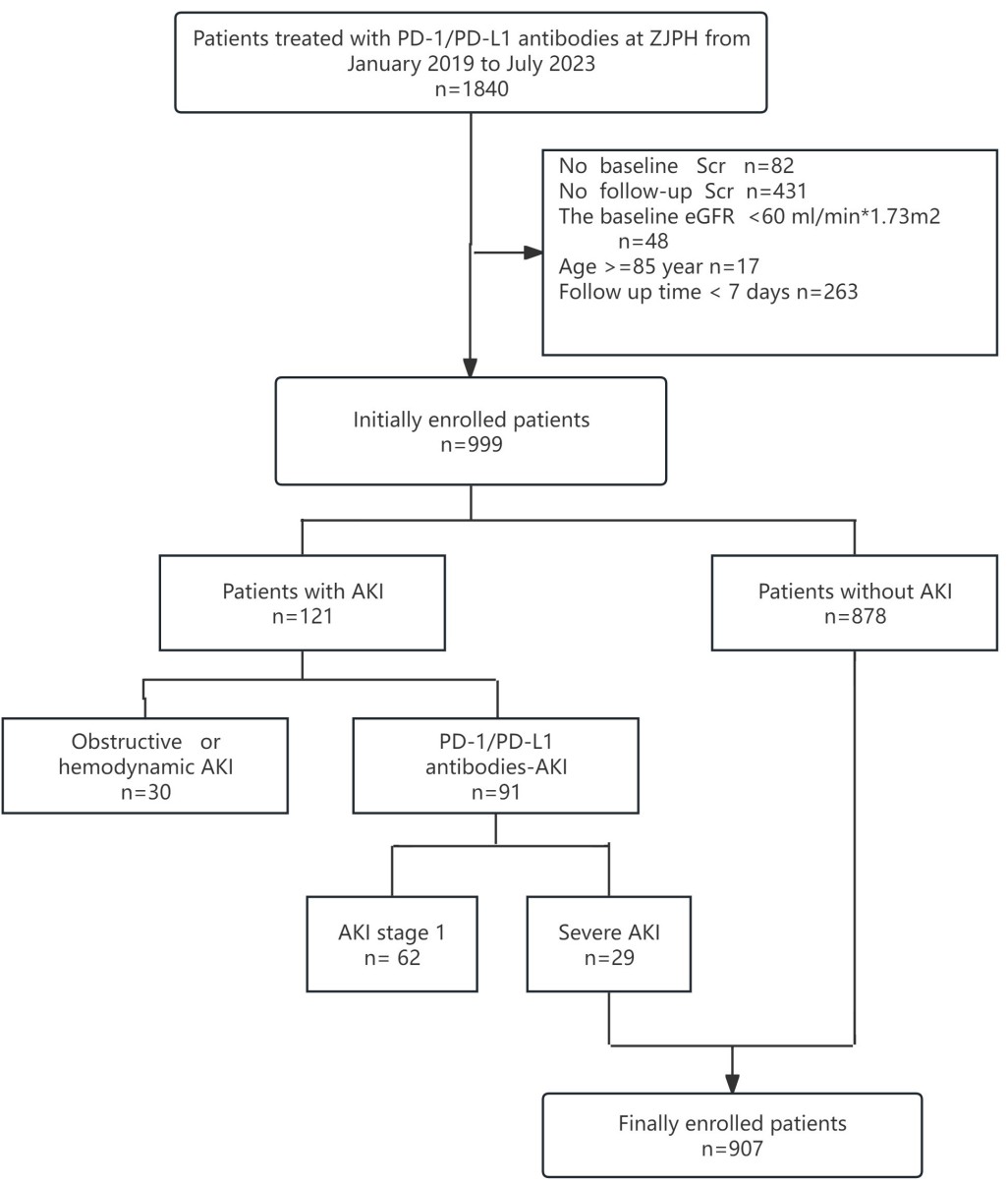

**Figure 1** **Flow diagram of the study design.** PD-1, programmed death-1; PD-L1, programmed death ligand-1; AKI, acute kidney injury; ZJPH, Zhejiang Provincial People's Hospital; Scr, serum creatinine; eGFR, estimated Glomerular Filtration Rate.

The Kaplan–Meier survival analysis showed significantly lower survival in patients with severe AKI (Fig. 2).

Lung cancer was the most prevalent tumor type, accounting for 304 cases (33.5%), followed by gastrointestinal tumors in 249 cases (27.5%) and hepatobiliary and pancreatic tumors in 205 cases (22.6%). Chemotherapy was administered to 673 patients (74.2%), with platinum-based agents being the most commonly used (316 patients, 34.8%), followed by gemcitabine (79 patients, 8.7%) and pemetrexed (56 patients, 6.2%). Over the follow-up
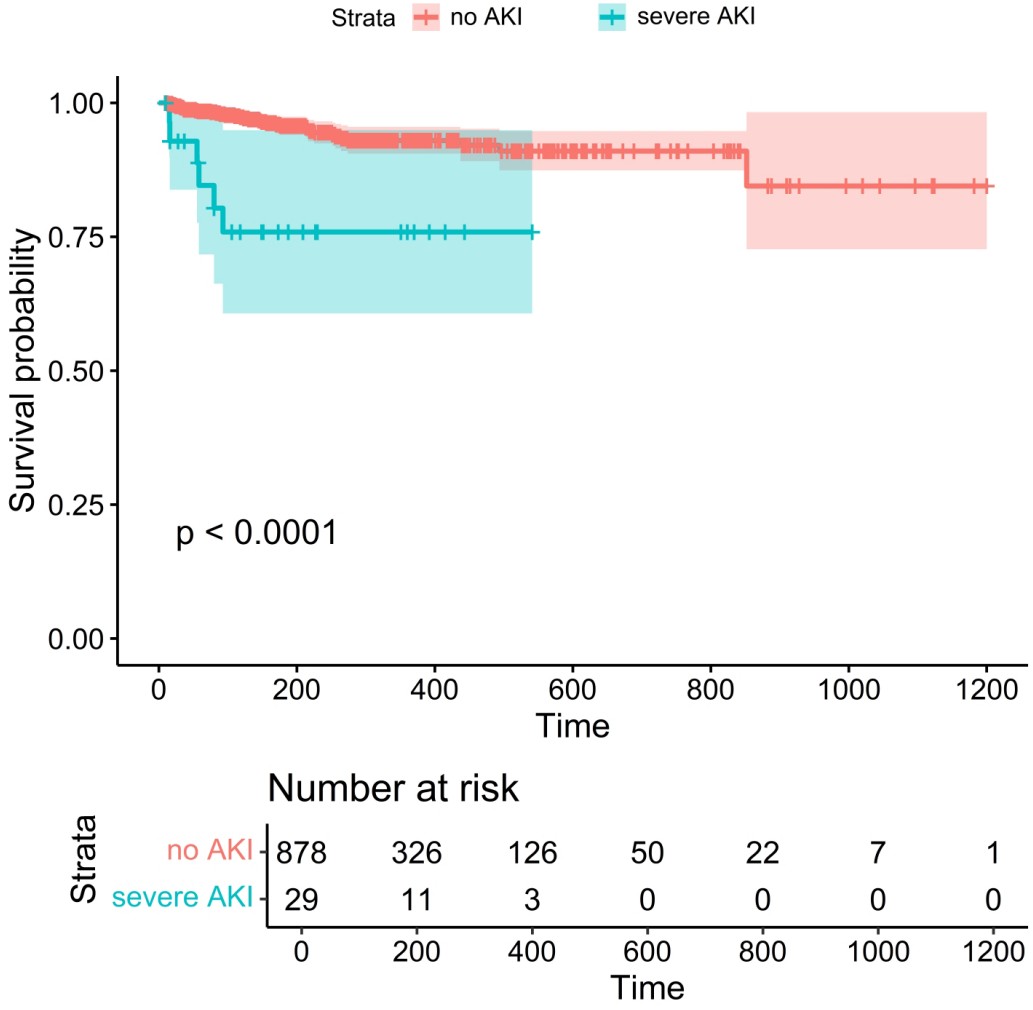

**Figure 2** Kaplan–Meier survival analysis showed that patients with severe AKI had a lower survival and poorer prognosis than those in the non-AKI group.

period, 42 patients (4.6%) died, including 20.7% (six patients) in the severe AKI group (Table 1).

No statistically significant differences were observed between the non-AKI and severe AKI groups regarding sex, age, smoking status and alcohol consumption, or comorbidities, including diabetes mellitus, hypertension, cirrhosis, chronic obstructive pulmonary disease (COPD), and cardiovascular disease ($p > 0.05$). Additionally, there were no significant differences between the groups in the occurrence of extrarenal iRAEs, with the exception of dysfunction. In the severe AKI group, thyroid dysfunction was present in six patients (20.7%). Among tumor types, only gastrointestinal tumors showed a significant difference between groups, with 6.9% in the severe AKI group compared to 28.1% in the non-AKI group ($p = 0.021$). Differences were also noted in the use of certain combined medications: ACEI/ARB use was more frequent in the severe AKI group (24.1% *vs.* 9.7%, $p = 0.026$),
**Table 1** Baseline characteristics between severe AKI and non-AKI cohorts.

| | Level | Overall | No-AKI | Severe AKI | *p* value |
|---|---|---|---|---|---|
| No. | | 907 | 878 | 29 | NA |
| Sex (%) | Male | 660 (72.8) | 643 (73.2) | 17 (58.6) | 0.127 |
| | Female | 247 (27.2) | 235 (26.8) | 12 (41.4) | |
| Age (years) | | 64 [57, 72] | 64 [57, 72] | 70 [61, 75] | 0.052 |
| Smoking (%) | | 276 (30.4) | 270 (30.8) | 6 (20.7) | 0.34 |
| Alcohol (%) | | 193 (21.3) | 188 (21.4) | 5 (17.2) | 0.757 |
| SBP (mmHg) | | 123 [113, 135] | 123 [113, 135] | 122 [114, 135] | 0.845 |
| DBP (mmHg) | | 75 [68, 82] | 75 [68, 82] | 75 [66, 84] | 0.835 |
| HBP (%) | | 277 (30.5) | 263 (30.0) | 14 (48.3) | 0.057 |
| DM (%) | | 134 (14.8) | 129 (14.7) | 5 (17.2) | 0.909 |
| COPD (%) | | 13 (1.4) | 12 (1.4) | 1 (3.4) | 0.893 |
| Cirrhosis (%) | | 70 (7.7) | 68 (7.7) | 2 (6.9) | 0.999 |
| CVD (%) | | 37 (4.1) | 36 (4.1) | 1 (3.4) | 0.999 |
| **AKI (%)** | Yes | 29 (3.2) | 0 (0.0) | 29 (100.0) | <0.001 |
| | No | 878 (96.8) | 878 (100.0) | 0 (0.0) | <0.001 |
| **AKI stage (%)** | Stage 2 | 23 (2.5) | 0 (0.0) | 23 (79.3) | |
| | Stage 3 | 6 (0.7) | 0 (0.0) | 6 (20.7) | |
| Follow up time (days) | | 140.8 [57.0, 294.2] | 138.9 [57.0, 294.5] | 148.9 [57.1, 229.9] | 0.691 |
| AKI occur time (days) | | NA | NA | 92.0 [25.0, 119.0] | NA |
| Died (%) | | 42 (4.6) | 36 (4.1) | 6 (20.7) | <0.001 |
| **Extra renal IRAEs (%)** | | 103 (11.4) | 96 (10.9) | 7 (24.1) | 0.056 |
| Thyroiddys function (%) | | 55 (6.1) | 49 (5.6) | 6 (20.7) | 0.003 |
| Skin injuries (%) | | 25 (2.8) | 24 (2.7) | 1 (3.4) | 0.999 |
| Gastrointestinal symptoms (%) | | 9 (1.0) | 9 (1.0) | 0 (0.0) | 0.999 |
| Fever (%) | | 30 (3.3) | 27 (3.1) | 3 (10.3) | 0.104 |
| Pneumonia (%) | | 8 (0.9) | 8 (0.9) | 0 (0.0) | 0.999 |
| Drug induced liver injury (%) | | 15 (1.7) | 15 (1.7) | 0 (0.0) | 0.999 |
| Other IRAEs (%) | | 5 (0.6) | 5 (0.6) | 0 (0.0) | 0.999 |
| **Tumor types** | | | | | |
| Hepatobiliary Pancreas (%) | | 205 (22.6) | 197 (22.4) | 8 (27.6) | 0.67 |
| Lung (%) | | 304 (33.5) | 296 (33.7) | 8 (27.6) | 0.626 |
| Urinary system (%) | | 53 (5.8) | 52 (5.9) | 1 (3.4) | 0.876 |
| gastrointestinal (%) | | 249 (27.5) | 247 (28.1) | 2 (6.9) | 0.021 |
| genitalsystmen (%) | | 51 (5.6) | 49 (5.6) | 2 (6.9) | 0.999 |
| Melanoma (%) | | 4 (0.4) | 4 (0.5) | 0 (0.0) | 0.999 |
| Head Neck (%) | | 43 (4.7) | 39 (4.4) | 4 (13.8) | 0.059 |
| Breast (%) | | 18 (2.0) | 18 (2.1) | 0 (0.0) | 0.919 |
| Other (%) | | 21 (2.3) | 19 (2.2) | 2 (6.9) | 0.298 |
| **Combined medications** | | | | | |
| PPI (%) | | 798 (88.0) | 771 (87.8) | 27 (93.1) | 0.567 |
| ACEI/ARB (%) | | 92 (10.1) | 85 (9.7) | 7 (24.1) | 0.026 |

Table 1 (*continued*)

| | Level | Overall | No-AKI | Severe AKI | *p* value |
|---|---|---|---|---|---|
| NSAIDs (%) | | 493 (54.4) | 467 (53.2) | 26 (89.7) | <0.001 |
| H2 blockers (%) | | 31 (3.4) | 31 (3.5) | 0 (0.0) | 0.61 |
| Iohexol (%) | | 523 (57.7) | 509 (58.0) | 14 (48.3) | 0.396 |
| Antibiotics (%) | | 16 (1.8) | 16 (1.8) | 0 (0.0) | 0.987 |
| Diuretics (%) | | 240 (26.5) | 222 (25.3) | 18 (62.1) | <0.001 |
| Chemotherapy (%) | | 673 (74.2) | 648 (73.8) | 25 (86.2) | 0.198 |
| **Anti-tumor drug class** | | | | | |
| Other Anti-tumor drugs (%) | | 673 (74.2) | 648 (73.8) | 25 (86.2) | 0.198 |
| Platinum drugs (%) | | 316 (34.8) | 302 (34.4) | 14 (48.3) | 0.179 |
| Gemcitabine (%) | | 79 (8.7) | 75 (8.5) | 4 (13.8) | 0.514 |
| Pemetrexed (%) | | 56 (6.2) | 56 (6.4) | 0 (0.0) | 0.312 |
| Cytotoxic drugs (%) | | 10 (1.1) | 8 (0.9) | 2 (6.9) | 0.033 |
| Targeted drug (%) | | 193 (21.3) | 189 (21.5) | 4 (13.8) | 0.441 |
| Interleukin (%) | | 18 (2.0) | 17 (1.9) | 1 (3.4) | 0.999 |
| **Immune check point inhibitor class (%)** | | | | | |
| PD1 | | 876 (96.6) | 848 (96.6) | 28 (96.6) | |
| PDL1 | | 20 (2.2) | 19 (2.2) | 1 (3.4) | 0.751 |
| PD1+PDL1 | | 11 (1.2) | 11 (1.3) | 0 (0.0) | |
| **Baseline laboratory data** | | | | | |
| eGFR (median [IQR])ml/min/1.73 m$^2$ | | 103.1 [87.9, 121.8] | 103.1 [88.2, 121.4] | 107.6 [76.4, 125.0] | 0.668 |
| Uric Acid, umol/L | | 296 [241, 359] | 298 [243, 361] | 234 [187, 315] | 0.01 |
| ALT, U/L | | 19.0 [13.0, 30.0] | 19.0 [13.0, 30.0] | 21.0 [13.0, 37.0] | 0.432 |
| AST, U/L | | 24.0 [18.0, 33.0] | 24.0 [18.0, 33.0] | 25.0 [20.0, 35.0] | 0.65 |
| GGT, U/L | | 36.0 [21.0, 73.0] | 36.0 [21.0, 71.8] | 41.0 [25.0, 163.0] | 0.239 |
| LDLC, mmol/L | | 2.60 [2.06, 3.18] | 2.62 [2.08, 3.19] | 2.46 [1.83, 3.04] | 0.3 |
| TC, mmol/L | | 4.34 [3.70, 5.08] | 4.36 [3.70, 5.08] | 4.02 [3.70, 4.98] | 0.24 |
| Albumin, g/L | | 36.1 [33.1, 39.3] | 36.2 [33.2, 39.3] | 33.1 [30.7, 38.1] | 0.025 |
| Globulin, g/L | | 29.8 [26.7, 33.4] | 29.8 [26.7, 33.4] | 31.5 [27.0, 32.8] | 0.997 |
| Phosphorus, mmol/L | | 1.13 [1.00, 1.26] | 1.13 [1.00, 1.27] | 1.14 [1.05, 1.24] | 0.986 |
| Glucose, mmol/L | | 5.17 [4.66, 5.81] | 5.17 [4.65, 5.80] | 5.62 [5.00, 5.95] | 0.123 |
| LDH, mmol/L | | 191 [159, 233] | 191 [159, 231] | 184 [164, 271] | 0.534 |
| Monocyte, $\times 10^9$ /L | | 0.40 [0.30, 0.55] | 0.40 [0.30, 0.54] | 0.40 [0.30, 0.67] | 0.508 |
| Eosinophil, $\times 10^9$ /L | | 0.10 [0.05, 0.19] | 0.10 [0.05, 0.19] | 0.12 [0.04, 0.20] | 0.755 |
| White blood cell, $\times 10^9$ /L | | 5.96 [4.50, 7.66] | 5.94 [4.49, 7.63] | 6.93 [5.30, 9.43] | 0.057 |
| Lymphocyte, $\times 10^9$ /L | | 1.20 [0.84, 1.60] | 1.20 [0.84, 1.60] | 0.92 [0.72, 1.26] | 0.012 |
| Hemoglobin, g/dl | | 12.0 [10.7, 13.3] | 12.1 [10.7, 13.3] | 11.1 [9.8, 12.0] | 0.014 |
| hsCRP, mg/L | | 8.4 [2.8, 30.0] | 8.0 [2.8, 29.9] | 21.2 [12.2, 40.2] | 0.015 |
| SG Urine | | 1.02 [0.01, 0.02] | 1.02 [0.01, 0.02] | 1.02 [0.01, 0.02] | 0.256 |
| Urinary leukocytes (%) | | 153 (16.9) | 145 (16.5) | 8 (27.6) | 0.189 |

**Notes.**

SBP, body mass index; DBP, diastolic blood pressure; CKD, chronic kidney disease; HBP, high blood pressure; DM, diabetes mellitus; COPD, chronic obstructive pulmonary disease; CVD, cardiovascular disease; IRAEs, immune related adverse events; eGFR, estimated glomerular filtration rate; AKI, acute kidney injury; ALT, alanine transaminase; GGT, gamma-glutamyl transpeptidase; LDLC, low-density lipoprotein cholesterol; AST, Aspartate aminotransferase; TC, total cholesterol; LDH, lactate dehydrogenase; hsCRP, high sensitivity C reactive protein; WBCJurine, white blood cell in the urine; PPI, proton pump inhibitor; ACEI, angiotensin-converting enzyme inhibitors; ARB, angiotensin receptor blockers; NSAIDs, non-steroidal antiinflammatory drugs; H2 blockers, H2-receptor antagonists; SG urine, specific gravity of urine.

as were NSAIDs (89.7% *vs.* 53.2%, $p < 0.001$) and diuretics (62.1% *vs.* 25.3%, $p < 0.001$). Regarding antitumor drug use, ICI usage did not differ significantly between the groups ($p > 0.05$). However, cytotoxic drug use was more frequent in the severe AKI group (6.9% *vs.* 0.9%, $p = 0.033$). Baseline parameters, including eGFR, alanine aminotransferase, aspartate transaminase, total cholesterol, low-density lipoprotein cholesterol (LDL-C), serum inorganic phosphate, glucose, monocytes, eosinophils, and urinary specific gravity, showed no statistically significant differences between the two groups. In the severe AKI group, elevated C-reactive protein levels (21.2 *vs.* 8.0, $p = 0.015$) were observed, alongside decreased levels of hemoglobin (11.1 *vs.* 12.1, $p = 0.014$), lymphocytes (0.92 *vs.* 1.20, $p = 0.012$), uric acid (234 *vs.* 298, $p = 0.01$), and albumin (33.1 *vs.* 36.2, $p = 0.025$).

## Risk factors for severe AKI

The risk factors associated with severe AKI were further examined in these patients. Using LASSO regression analysis, 56 characteristic variables were included (Fig. S1A). This analysis identified 18 factors that were significantly associated with the occurrence of severe AKI (Fig. S1B & Table 2). These 18 factors included age, high blood pressure, diabetes mellitus, gastrointestinal tumor, head neck tumor, other tumors, gamma-glutamyl transferase (GGT), albumin, hemoglobin, white blood cell in the urine, ACEI/ARB use, NSAID use, iohexol use, diuretics, platinum drugs, pemetrexed, cytotoxic drugs, and proton pump inhibitor. These factors were then analyzed using univariate and multivariate Cox regression, which finally identified four independent risk factors for severe AKI: GGT (hazard ratio (HR): 1.17; 95% confidence interval CI [1.01–1.03], $p = 0.008$), NSAIDs (HR: 4.58; 95% CI [1.36–15.46], $p = 0.014$), diuretics (HR: 3.61; 95% CI [1.62–8.07], $p = 0.002$), and cytotoxic drugs (HR: 5.04; 95% CI [1.12–22.81], $p = 0.036$) (Fig. 3). The dynamic alterations of the concordance index of severe AKI are shown (Fig. 4).

When GGT > 250 U/L was used as the threshold, the Kaplan−Meier survival curve analysis showed a significantly higher incidence of severe AKI in patients exceeding this threshold ($p < 0.0001$) (Fig. 5A). Similarly, the incidence of severe AKI was notably elevated among patients receiving diuretics compared to those who did not ($p < 0.0001$) (Fig. 5B). Additionally, NSAID use was associated with higher severe AKI incidence than non-use ($p = 0.00023$) (Fig. 5C). Lastly, severe AKI incidence was significantly higher in patients treated with cytotoxic drugs than in those not receiving these medications ($p < 0.0001$) (Fig. 5D).

## Prognosis of patients

The 29 patients diagnosed with severe AKI were monitored for an average duration of 92 days. Among these patients, two achieved full recovery, nine experienced partial recovery, 13 did not recover, and five were lost to follow-up after being diagnosed with severe AKI. In the subgroup of 23 patients with stage 2 AKI, two recovered fully, seven had partial recovery, 10 did not recover, and the remaining four were lost to follow-up. Of the six patients with stage 3 AKI, two recovered partially, three did not recover, and one was lost to follow-up (Fig. 6).

**Table 2 Risk factors of severe AKI after PD1/PD L1 therapy.**

| Variables | Univariate analysis | | Multivariate analysis | |
|---|---|---|---|---|
| | **P** | **HR (95%CI)** | **P** | **HR (95%CI)** |
| Age | 0.037 | 1.04 (1.01 ∼ 1.08) | 0.054 | 1.04 (1.00 ∼ 1.07) |
| HBP | | | | |
| 0 | | 1.00 (Reference) | | |
| 1 | 0.057 | 2.03 (0.98 ∼ 4.20) | | |
| DM | | | | |
| 0 | | 1.00 (Reference) | | |
| 1 | 0.799 | 1.13 (0.43 ∼ 2.97) | | |
| Gastrointestinal tumor | | | | |
| 0 | | 1.00 (Reference) | | 1.00 (Reference) |
| 1 | 0.021 | 0.18 (0.04 ∼ 0.77) | 0.081 | 0.27 (0.06 ∼ 1.17) |
| Head Neck tumor | | | | |
| 0 | | 1.00 (Reference) | | |
| 1 | 0.063 | 2.73 (0.95 ∼ 7.84) | | |
| Other tumors | | | | |
| 0 | | 1.00 (Reference) | | |
| 1 | 0.056 | 4.07 (0.97 ∼ 17.16) | | |
| GGT, U/L/10 | <.001 | 1.02 (1.01 ∼ 1.03) | 0.008 | 1.02 (1.01 ∼ 1.03) |
| Albumin, g/L | 0.001 | 0.88 (0.81 ∼ 0.95) | 0.326 | 0.96 (0.87 ∼ 1.05) |
| Hemoglobin, g/dl | 0.006 | 0.78 (0.65 ∼ 0.93) | 0.233 | 0.88 (0.71 ∼ 1.09) |
| WBCUrine | | | | |
| 0 | | 1.00 (Reference) | | |
| 1 | 0.125 | 1.89 (0.84 ∼ 4.27) | | |
| ACEI/ARB | | | | |
| 0 | | 1.00 (Reference) | | 1.00 (Reference) |
| 1 | 0.029 | 2.58 (1.10 ∼ 6.03) | 0.609 | 1.28 (0.49 ∼ 3.35) |
| NSAIDs | | | | |
| 0 | | 1.00 (Reference) | | 1.00 (Reference) |
| 1 | 0.002 | 6.91 (2.09 ∼ 22.84) | 0.014 | 4.58 (1.36 ∼ 15.46) |
| Iohexol | | | | |
| 0 | | 1.00 (Reference) | | 1.00 (Reference) |
| 1 | 0.046 | 0.47 (0.23 ∼ 0.99) | 0.160 | 0.58 (0.27 ∼ 1.24) |
| Diuretics | | | | |
| 0 | | 1.00 (Reference) | | 1.00 (Reference) |
| 1 | <.001 | 5.06 (2.39 ∼ 10.72) | 0.002 | 3.61 (1.62 ∼ 8.07) |
| Platinum drugs | | | | |
| 0 | | 1.00 (Reference) | | |
| 1 | 0.147 | 1.71 (0.83 ∼ 3.55) | | |
| Pemetrexed | | | | |
| 0 | | 1.00 (Reference) | | |
| 1 | 0.996 | 0.00 (0.00 ∼ Inf) | | |

**Table 2** (*continued*)

| Variables | Univariate analysis | | Multivariate analysis | |
|---|---|---|---|---|
| | *P* | HR (95%CI) | *P* | HR (95%CI) |
| Cytotoxic drugs | | | | |
| 0 | | 1.00 (Reference) | | 1.00 (Reference) |
| 1 | 0.001 | 10.44 (2.47 ∼ 44.17) | 0.036 | 5.04 (1.12 ∼ 22.81) |
| PPI | | | | |
| 0 | | 1.00 (Reference) | | |
| 1 | 0.532 | 1.58 (0.38 ∼ 6.65) | | |

**Notes.**

HR, Hazard Ratio; CI, Confidence Interval; HBP, high blood pressure; DM, diabetes mellitus; GGT, gamma-glutamyl transpeptidase; WBCJ urine, white blood cell in the urine; PPI, proton pump inhibitor; ACEI, angiotensin-converting enzyme inhibitors; ARB, angiotensin receptor blockers; NSAIDs, non-steroidal antiinflammatory drugs; PPI, proton pump inhibitor.

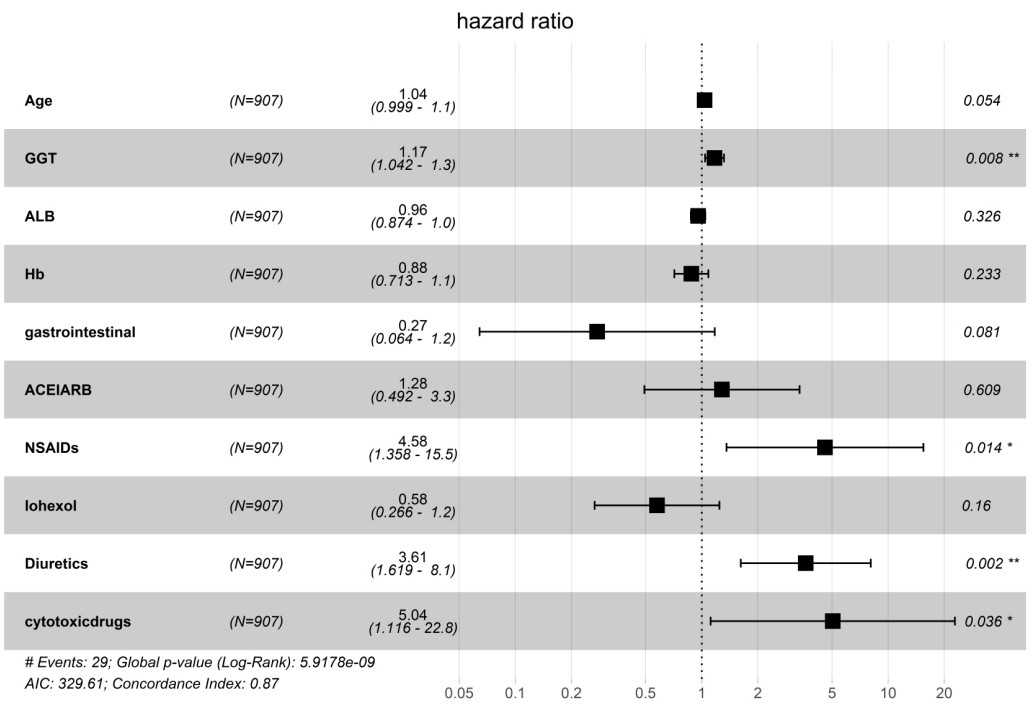

**Figure 3** **Multivariate COX regression analysis of independent risk factors associated with severe AKI.** GGT, gamma-glutamyl transpeptidase; ALB, albumin; Hb, hemoglobin; ACEI, angiotensin-converting enzyme inhibitor; ARB, angiotensin receptor blocker; NSAIDs, nonsteroidal anti -inflammatory drug.

## DISCUSSION

Recent studies suggest that AKI significantly impacts the prognosis of patients undergoing ICI therapy, such as PD-1/PD-L1 inhibitors (*Shingarev & Glezerman, 2019*; *Cortazar et al., 2020*; *Venkatachalam et al., 2020*; *García-Carro et al., 2022*; *Sprangers et al., 2022*). Evidence shows that failure to recover kidney function in these patients can lead to the permanent discontinuation of ICI treatment, progression to chronic kidney failure, and an increased

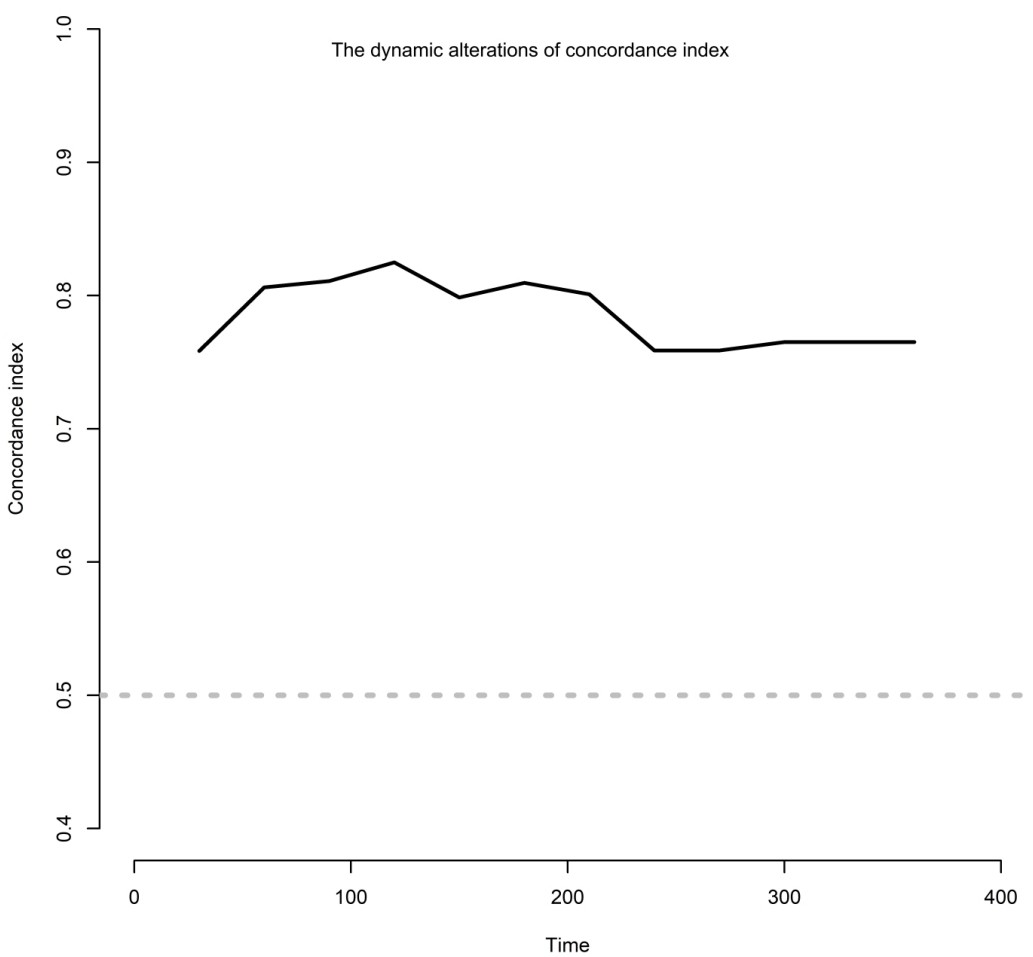

**Figure 4  The dynamic alterations of concordance index.**

risk of mortality (*Moturi, Sharma & Hashemi-Sadraei, 2023*). This retrospective study evaluated 907 patients with various tumors who received at least one dose of PD-1/PD-L1 antibody therapy. During the follow-up, 29 patients experienced severe AKI, with an incidence rate of 3.2%. The median time for severe AKI onset after starting PD-1/PD-L1 therapy was 92 days.

Patients who developed severe AKI exhibited a higher mortality rate, which underscored the critical need for active monitoring and management of renal function in patients with cancer treated with PD-1/PD-L1 antibodies. Identifying and understanding the associated risk factors is also essential. In this study, four risk factors associated with severe AKI were identified in patients undergoing therapy with PD-1/PD-L1 antibodies: GGT level, NSAID use, diuretic use, and cytotoxic drug use. Time-dependent C-index analyses were used to evaluate the model's discrimination ability for predicting severe AKI, demonstrating strong predictive performance and confirming its accuracy and clinical applicability.

The methodology employed in this study differs from traditional statistical approaches, such as logistic regression or risk factor selection based on clinical experience (*Meraz-Muñoz*

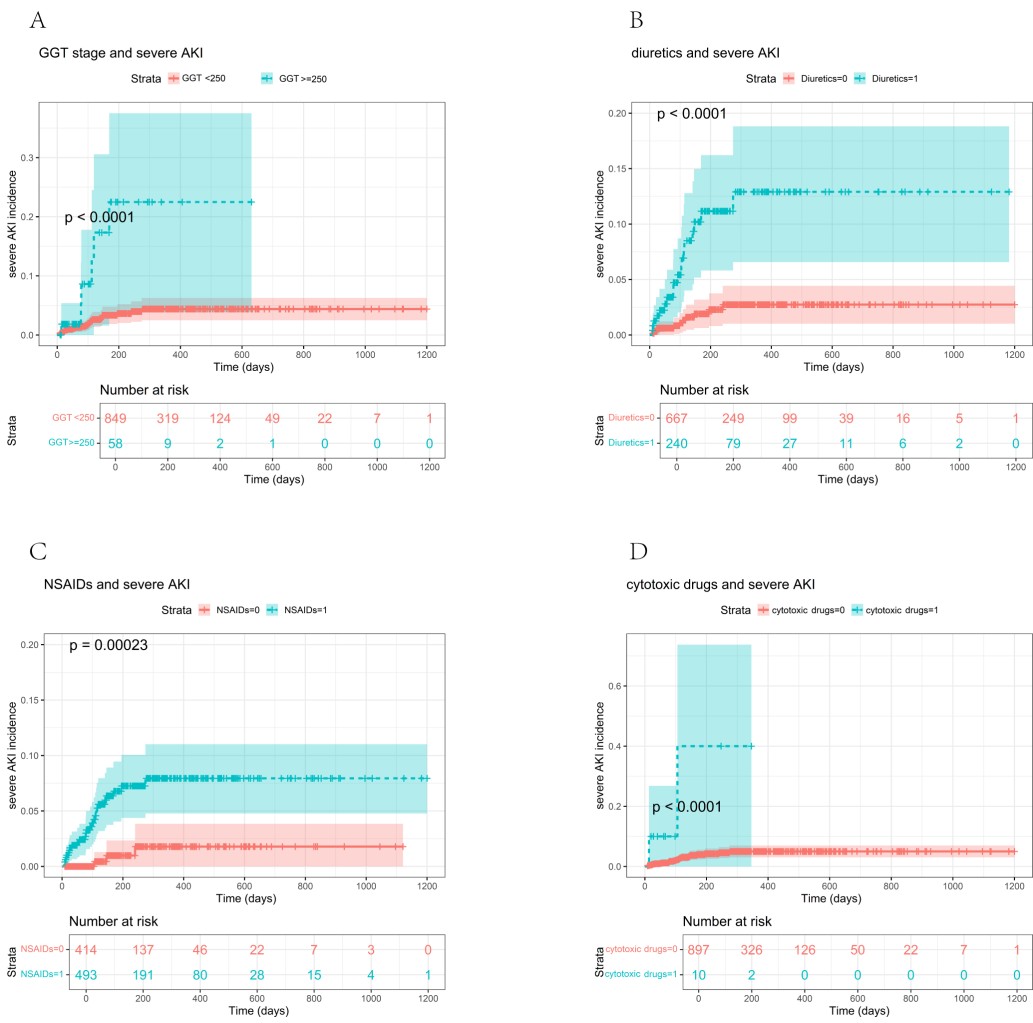

**Figure 5 Relationship between the four variables and severe AKI incidence.** (A) Severe AKI incidence according to GGT ($p < 0.0001$); (B) Severe AKI incidence according to diuretic medicine ($p < 0.0001$); (C) Severe AKI incidence according to NSAIDs ($p < 0.001$); (D) Severe AKI incidence according to cytotoxic drugs ($p < 0.0001$).

*et al., 2020*; *Ji et al., 2022*). Given the small number of positive AKI cases relative to the 77 variables analyzed, additional statistical methods were required for accurate feature selection. Thus, LASSO regression analysis was chosen as a screening method, allowing for a more robust handling of complex datasets. By constructing penalty functions to compress variable coefficients, LASSO regression effectively mitigates the risk of overfitting, yielding the minimum value for the absolute sum of the regression coefficients and the residual sum under specified constraints. This method selectively reduces some regression coefficients to zero, enabling the efficient analysis of numerous clinical factors and variables (*Wu et al., 2022*).

Diuretics can reduce renal blood flow and the glomerular filtration rate both directly and indirectly by modulating the tubular feedback mechanism, leading to renal ischemia,

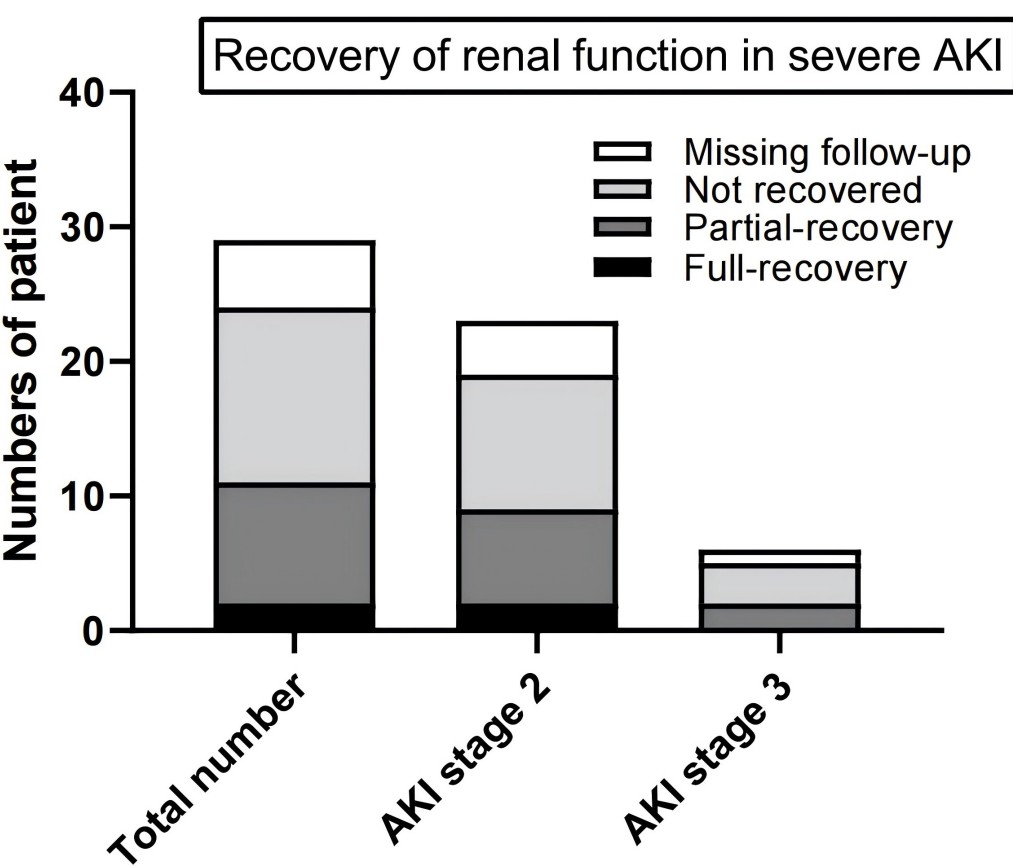

**Figure 6** **Recovery of renal function of patients with severe AKI.** Of the 29 patients with severe AKI, two fully recovered, nine partially recovered, 13 did not recover, and five were lost to follow-up. Among the 23 stage 2 AKI patients, two fully recovered, seven partially recovered, 10 did not recover, and four were lost to follow-up. Of the six stage 3 AKI patients, two partially recovered, three did not recover, and one was lost to follow-up.

hypoxia, and hypoperfusion, all of which impair renal function (*Perazella & Rosner, 2022*). Additionally, PD-1/PD-L1 antibodies can activate autoreactive T cells, resulting in renal tubular epithelial cell dysfunction (*Liu et al., 2023a*; *Liu et al., 2023a*; *Liu et al., 2023b*; *Moturi, Sharma & Hashemi-Sadraei, 2023*). Thus, diuretic use may elevate the risk of kidney damage in patients receiving these therapies (*Ji et al., 2022*). Our data analysis also indicated that elevated GGT levels contribute to severe AKI, a finding not commonly reported in prior studies. GGT, a liver enzyme involved in transporting glutathione and amino acids into cells, plays a role in glutathione metabolism. Although widely distributed, GGT is found in particularly high concentrations in the kidneys (*Cutrín et al., 2000*). Previous research has suggested that urinary GGT levels may serve as a marker of renal tubular damage (*Bagshaw et al., 2007*; *Nejat et al., 2012*; *Marcelino et al., 2014*; *Peng et al., 2024*). By raising intracellular glutathione levels, GGT may influence T cell response to oxidative stress (*Carlisle, King & Karp, 2003*), support reactive oxygen species balance in tumor cells, and protect tumor cells from treatment-related cell death through the redox

pathway (*Zhang & Forman, 2009*). Moreover, time-dependent C-index analyses were used to evaluate the model's discrimination ability for predicting severe AKI, demonstrating strong predictive performance and confirming its accuracy and clinical applicability nephropathy (*Javid et al., 2023*). We speculate that the activation of glutathione increases the nephrotoxicity of PD-1/PD-L1 antibodies, leading to the development of severe AKI.

The use of certain cytotoxic drugs in patients receiving PD-1/PD-L1 antibody therapy was linked to a heightened risk of severe AKI. This may be attributed to the inherent nephrotoxicity of these drugs, which stems from their antitumor mechanisms. Cytotoxic drugs often inhibit DNA repair and promote oxidative stress, which directly affects proximal renal tubule cells, aligning with findings from previous studies (*Perazella & Rosner, 2022*; *Liu et al., 2023a*; *Liu et al., 2023b*; *Lou et al., 2023*).

NSAID use was also identified as an independent risk factor for severe AKI in this study. NSAIDs can induce acute tubulointerstitial nephritis (ATIN), potentially due to T-cell activation associated with hypersensitivity reactions (*Liu et al., 2023a*; *Liu et al., 2023b*; *Sanchez-Alamo, Cases-Corona & Fernandez-Juarez, 2023*). Furthermore, the use of PD-1/PD-L1 antibodies may lower the patients' tolerance thresholds by activating drug-specific T cells, which could lead to a loss of tolerance to potentially nephrotoxic drugs (*Ji et al., 2022*; *Perazella & Rosner, 2022*).

PPIs are commonly used in antitumor therapies, and many studies have suggested they may increase the risk of AKI (*Seethapathy et al., 2019*; *Gupta et al., 2021*; *Koks et al., 2021*; *Gérard et al., 2022*), potentially due to effector T-cell activation following T-cell sensitization with PD-1/PD-L1 antibodies. However, a recent large-scale cohort study found that, after adjusting for various confounding factors, the association between proton pump inhibitor (PPI) use and AKI was significantly reduced (*Munch et al., 2024*). In the present study, no association was observed between PPI use and the risk of severe AKI, suggesting that PPI use may be linked primarily to mild forms of AKI.

Upon reviewing the outcomes for the 29 patients diagnosed with severe AKI, it was observed that more than half of these individuals failed to regain their kidney function. This non-recovery rate is notably higher than what has been reported in previous studies (*Cortazar et al., 2020*; *Gupta et al., 2021*; *Koks et al., 2021*). The recovery of renal function in patients with cancer treated with PD-1/PD-L1 antibodies significantly affects subsequent treatment options (*Cortazar et al., 2020*). The low recovery rate observed in this study led us to consider potential contributing factors, including the level of attention oncologists pay to severe AKI during treatment, as well as the degree of kidney specialist involvement in the biopsy, treatment, and follow-up processes.

It is essential to acknowledge the limitations present in this study. First, the retrospective nature of its design could have led to selection and information biases owing to the reliance on medical records, which may not be complete. Second, as a single-center study, the findings may have limited generalizability. Although the sample size was relatively large (907 patients), it may still be insufficient to identify all possible risk factors for severe AKI in those patients. Furthermore, the lack of a renal biopsy in most cases limited our ability to confirm AKI etiology, and we may have included cases of AKI unrelated to anti-PD-1/PD-L1 antibody-induced ATIN. Future research should incorporate multicenter

data, larger and more diverse cohorts, and extended follow-up periods to validate these findings and reduce potential biases.

## CONCLUSIONS

A 3.2% incidence of severe AKI was observed among patients undergoing PD-1/PD-L1 therapy, with a mortality rate of 20.7%. Key risk factors identified were elevated GGT levels and the use of NSAIDs, diuretics, and cytotoxic drugs. These findings offer valuable data for clinicians and researchers, highlighting the need for vigilant monitoring and early identification of high-risk patients on PD-1/PD-L1 inhibitors. Increased awareness and timely intervention can enhance patient outcomes, improve survival rates, and help maintain the continuity of cancer treatment. Future studies with larger randomized controlled trials are recommended to validate these risk factors further and provide clinical guidance on managing AKI risks in patients receiving PD-1/PD-L1 therapies.

## ACKNOWLEDGEMENTS

We would like to thank our colleagues at Zhejiang Provincial People's Hospital for their valuable contributions to this study. We are grateful for the technical support and data platform from Yidu Cloud Technology Company Ltd. Lastly, we would like to thank Editage for English language editing.

### Funding

This work was supported by the National Natural Science Foundation of China (No. 82202042), Zhejiang Provincial Natural Science Foundation of China under (Grant No. LY24H050002), and the Zhejiang Medical and Health Science and Technology Program (NO. 2025KY580). The funders had no role in study design, data collection and analysis, decision to publish, or preparation of the manuscript.

### Grant Disclosures

The following grant information was disclosed by the authors:
National Natural Science Foundation of China: 82202042.
Zhejiang Provincial Natural Science Foundation of China: LY24H050002.
Zhejiang Medical and Health Science and Technology Program: 2025KY580.

### Competing Interests

The authors declare there are no competing interests.

### Author Contributions

- Yuemeng Wu conceived and designed the experiments, performed the experiments, analyzed the data, prepared figures and/or tables, and approved the final draft.
- Lingfan Luo performed the experiments, analyzed the data, prepared figures and/or tables, and approved the final draft.

- Xin Sun performed the experiments, authored or reviewed drafts of the article, and approved the final draft.
- Xiaolan Ye performed the experiments, authored or reviewed drafts of the article, and approved the final draft.
- Yan Ren conceived and designed the experiments, analyzed the data, authored or reviewed drafts of the article, and approved the final draft.
- Wei Zhang conceived and designed the experiments, prepared figures and/or tables, and approved the final draft.
- Shuangshan Bu analyzed the data, authored or reviewed drafts of the article, and approved the final draft.
- Yiwen Li conceived and designed the experiments, authored or reviewed drafts of the article, and approved the final draft.
- Bin Zhu conceived and designed the experiments, prepared figures and/or tables, authored or reviewed drafts of the article, and approved the final draft.
- Lina Shao conceived and designed the experiments, analyzed the data, prepared figures and/or tables, authored or reviewed drafts of the article, and approved the final draft.

## Human Ethics

The following information was supplied relating to ethical approvals (i.e., approving body and any reference numbers):

This study was approved by the Ethics Research Committee of the Zhejiang Provincial People's Hospital (Approval number: QT2024180).

## Data Availability

Code and raw data are available in the Supplemental Files.

## Supplemental Information

Supplemental information for this article can be found online at http://dx.doi.org/10.7717/peerj.19886#supplemental-information.

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
