# Peer review of "A closer look at severe acute kidney injury: risk factors and outcomes in PD-1/PD-L1 antibody treatment from a retrospective study"

_PeerJ, doi:10.7717/peerj.19886_

## Round 0.1 · original submission · Minor Revisions

In addition to the comments by the authors, please deposit your code and raw data in Zenodo or another appropriate database with a DOI. Additionally, statistical packages used (including R packages) should be cited with the appropriate versions and references.

Reviewer 1 ·

Basic reporting

In the end of Introduction, it should briefly cover the main findings of the study.

The abstract needs to be more concise without the unnecessary details such as characteristics of the patients, CI ranges, p-values, date, definition of AKI, etc. It only needs to briefly introduce the background, report the number of patients this study included, methods used (multivariate Cox regression), and the findings (what are the significant risk factors).

Experimental design

The Method section mentions that “Imputation was performed for missing data when the proportion of missing values was less than 20%”. First, it did not describe how the imputation was performed. Second, it did not justify why 20% is an acceptable cutting point. It seems to me that, if a variable (let’s say, age, weight, etc.) is missing in 20% of the patients, it is simply not acceptable to “guess” those variables for up to 20% of the patients by, for example, taking the average of all the patients.

According to the Method section, the authors employed two variable selection procedures: LASSO regression and univariate Cox regression. The selected variables are then subjected to multivariable Cox regression. The variable selection is the major issue in this study.

First, the authors did not justify why they have to perform variable selection in the first place. According to the results, they selected 18 risk factors out of 56—which does not seem that many to start with. Besides, pre-selecting variables may introduce bias to the final multivariate Cox regression.

Second, the authors did not sufficiently describe how they perform LASSO regression: what is the response variable? How did they set the lambda parameter?

Third, the authors did not justify why they used two different variable selection methods, instead of either one of them.

Validity of the findings

Figure 3 is not relevant to the findings—it shows details of LASSO regression and does not deliver any meaningful message.

Figure 6 is kind of redundant because the tables already cover those messages.

Figure 5 and the corresponding text about concordance index does not contribute to any discussion or deliver any findings.

Reviewer 2 ·

Basic reporting

No comment

Experimental design

Retrospective study included 907 patients. Doing LASSO for the statistical analysis-LASSO regression is well-suited for this study as it helps identify the most important predictors while preventing overfitting, especially in high-dimensional data

Validity of the findings

The author classifies both Stage 2 and Stage 3 AKI as severe AKI; however, Stage 2 is considered moderate AKI. Based on this classification, only six patients fall into the severe AKI category, which may compromise the study's validity.
Rather than calling it as severe AKI, moderate to severe AKI

Additional comments

1. Serum inorganic phosphate" is the correct term instead of "blood phosphorus.
2. What could explain the lower serum uric acid levels in severe AKI?
3. However, eGFR values over 90 mL/min/1.73m² are often rounded and reported as "eGFR >90
4.Sentence 294 -Upon reviewing the outcomes for the 29 patients diagnosed with severe AKI, it was observed that more than half of these individuals failed to regain their kidney function. This recovery rate is notably higher than what has been reported in previous studies- it sounds contradictory. If more than half failed to regain kidney function, that suggests a low recovery rate, yet the next sentence implies a higher recovery rate than previous studies.
5. Authors can quote this article "Ann Med Surg (Lond). 2023 Jun 20;85(8):4033–4040. doi: 10.1097/MS9.0000000000000967" Serum GGT is elevated in contrast induced nephropathy.

---

## Round 0.2 · Minor Revisions

Please address the eGFR comment by Reviewer #2 and please make your code comments be in English per the journal's standard. Thank you!

Reviewer 1 ·

Basic reporting

The reviewers have addressed my previous comments.

Experimental design

The reviewers have addressed my previous comments.

Validity of the findings

The reviewers have addressed my previous comments.

Reviewer 2 ·

Basic reporting

No comment

Experimental design

No comment

Validity of the findings

No comment

Additional comments

However, eGFR values over 90 mL/min/1.73m² are often rounded and reported as "eGFR >90
Response: Thank you for your comment. Apologies, we were unable to locate the part you are referring to.
Sorry for not being clear in my report. In the table, eGFR values have been reported >90 ml/min/1.73m2.
Is it a practise to report eGFR > 90 as the actual number rather than as > 90.

---

## Round 0.3 · Minor Revisions

Hello,

Will you please clarify if you are unable to present the actual eGFR values, and if so, why is that?

Thank you.

---

## Round 0.4 · accepted · Accept

Thank you for addressing all of the reviewer comments. This is now ready for publication.